# Targeting Feedforward Loops Formed by Nuclear Receptor RORγ and Kinase PBK in mCRPC with Hyperactive AR Signaling

**DOI:** 10.3390/cancers13071672

**Published:** 2021-04-01

**Authors:** Xiong Zhang, Zenghong Huang, Junjian Wang, Zhao Ma, Joy Yang, Eva Corey, Christopher P. Evans, Ai-Ming Yu, Hong-Wu Chen

**Affiliations:** 1Department of Biochemistry and Molecular Medicine, School of Medicine, University of California, Davis, Sacramento, CA 95817, USA; lqqzhang@ucdavis.edu (X.Z.); znhhuang@ucdavis.edu (Z.H.); wangjj87@mail.sysu.edu.cn (J.W.); mzma@ucdavis.edu (Z.M.); aimyu@ucdavis.edu (A.-M.Y.); 2Department of Urology, School of Medicine, University of California, Davis, Sacramento, CA 95817, USA; jcyang@ucdavis.edu (J.Y.); cpevans@ucdavis.edu (C.P.E.); 3Department of Urologic Surgery, University of Washington, Seattle, WA 98195, USA; ecorey@uw.edu; 4Comprehensive Cancer Center, University of California, Davis, Sacramento, CA 95817, USA; 5VA Northern California Health Care System-Mather, Mather, CA 95655, USA

**Keywords:** CRPC, RORγ, antagonists, inverse agonists, AR, AR-V7, PBK, kinase, PDX, cell invasion, EMT, metastasis

## Abstract

**Simple Summary:**

Prostate cancer is one of the most frequently diagnosed cancers in men and is the second leading cause of cancer death in developed countries. Current therapeutics that target the androgen receptor (AR) are only transiently effective. Anti-AR therapy-resistant tumors often emerge with vast cellular and molecular alterations and present themselves at the clinic in more deadly forms including metastatic castration-resistant prostate cancer (mCRPC) or neuroendocrine prostate cancer (NEPC). One emerging strategy in effective treatment of the advanced forms of prostate cancer is to target drivers other than AR. The present study shows that the nuclear receptor RORγ and the serine/threonine kinase PBK form a regulatory loop in hyperactive AR signaling. It also demonstrates that orally administered, small-molecule antagonists/inverse agonists of RORγ are effective in blocking the growth of the mCRPC tumors. Our findings provide a rationale for therapeutic targeting of RORγ alone or in combination with PBK inhibitors for the advanced forms of prostate cancer.

**Abstract:**

Metastatic castration-resistant prostate cancer (mCRPC) is a highly aggressive disease with few therapeutic options. Hyperactive androgen receptor (AR) signaling plays a key role in CRPC progression. Previously, we identified RAR-related orphan receptor gamma (RORγ) as a novel key driver of AR gene overexpression and increased AR signaling. We report here that several RORγ antagonists/inverse agonists including XY018 and compound 31 were orally effective in potent inhibition of the growth of tumor models including patient-derived xenograft (PDX) tumors. RORγ controls the expression of multiple aggressive-tumor gene programs including those of epithelial-mesenchymal transition (EMT) and invasion. We found that PDZ binding kinase (PBK), a serine/threonine kinase, is a downstream target of RORγ that exerts the cellular effects. Alterations of RORγ expression or function significantly downregulated the mRNA and protein level of PBK. Our further analyses demonstrated that elevated PBK associates with and stabilizes RORγ and AR proteins, thus constituting novel, interlocked feed-forward loops in hyperactive AR and RORγ signaling. Indeed, dual inhibition of RORγ and PBK synergistically inhibited the expression and function of RORγ, AR, and AR-V7, and the growth and survival of CRPC cells. Therefore, our study provided a promising, new strategy for treatment of advanced forms of prostate cancer.

## 1. Introduction

The hyperactivity of androgen receptor (AR), a member of the nuclear receptor (NR) family of transcription factors, is the major driver of prostate cancer (PCa) progression to metastatic castration-resistant PCa or mCRPC. Mechanisms contributing to AR hyperactivity in the tumors include AR gene overexpression, amplification, mutations, and aberrant alternative splicing that generates constitutively active AR variants, such as AR-V7, that lack the ligand binding domain (LBD) [1,2,3]. Other major mechanisms such as aberrant kinase signaling and protein-protein interactions to stimulate AR activities have also been documented. Among the kinase signaling pathways important for mCRPC are the interplays between Pten-loss/PI3K/AKT, Src and AR [4,5,6,7,8]. Notably, several kinases—including the serine/threonine PDZ binding kinase (PBK)—were shown to regulate AR stability [9]. Of particular interest is that PBK is an AR target which is overexpressed in subsets of aggressive PCa tumors and that its overexpression correlates with poor clinical outcome [9].

Currently, therapeutics suppressing AR activity are the mainstay of systemic therapy for advanced PCa. Widely used AR antagonists such as bicalutamide and enzalutamide targeting the AR LBD exhibit initial favorable response in most patients. However, resistance to those drugs inevitably develops and the disease relapses through tumor genetic and epigenetic changes, including the generation of AR variants such as AR-V7 [1,3,10,11]. Currently, therapeutics are being developed to target the AR variants while non-AR targeting therapeutic strategies are also being actively pursued [1,12,13,14,15,16].

In our pursuit of developing new therapeutics for advanced PCa, we identified retinoic acid receptor-related orphan receptor-γ (RORγ), another member of the NR family [17], as one of the major upstream regulators of the aberrant AR expression and function in CRPC [18]. RORγ, encoded by RORC gene, and its sub-family member RORα and RORβ act primarily as transcriptional activators in control of gene programs important for circadian rhythm, metabolism and development [19]. T cell-specific isoform RORγt plays a crucial role in T cell differentiation to Th17 which is important for immunity and autoimmune disease development. Recent studies demonstrate that certain oxysterols and intermediates of cholesterol biosynthesis can serve as endogenous RORγ activating ligands/agonists [19]. We previously reported a series of RORγ antagonists/inverse agonists with different scaffolds [18,20,21,22,23]. Compounds with biphenyl-4-yl-amines and related scaffolds displayed potent anti-tumor activities in cell culture and in tumor models [18,22,23,24,25]. We showed that their major mechanism of action in PCa is potent inhibition of AR expression and function in CRPC cell and tumors.

In this study, we demonstrate that via oral dosing several RORγ antagonists/inverse agonists including compound 31 (cmpd 31 hereafter) are highly effective in inhibition of the growth of patient-derived xenograft (PDX) tumors. We found that RORγ controls the expression of multiple aggressive-tumor gene programs including epithelial-mesenchymal transition (EMT) genes such as PTTG and PBK. Interestingly, PBK associates with and stabilizes RORγ protein, thus constituting novel, interconnected feedforward loops in AR hyperactivity and a vulnerability point in effective therapeutic targeting for mCRPC.

## 2. Results

### 2.1. RORγ Antagonist/Inverse Agonist cmpd 31 Ootently Inhibits CRPC Cell Survival, Migration, and Invasion

We previously identified and tested a series of compounds with biphenyl-4-yl-amines scaffold that were derived initially by combining the structural features of SR2211 and the GSK compound (Figure 1A). The amide derivative compounds XY018 and XY101 displayed excellent potencies in antagonizing the transcriptional activity of RORγ and inhibition of the growth of prostate cancer and breast cancer tumors [18,22,24]. Interestingly, cmpd 31, which contains a 4,4,4-trifluorobutyryl group (Figure 1A), displayed a stronger RORγ LBD-stabilization activity than XY018 (with a temperature shift of 7.8 °C by cmpd 31 vs. 4.2 °C by XY018 in thermal shift assay) [22]. We thus examined the potency of cmpd 31 in suppression of cancer cell and tumor growth. Indeed, our cell viability assay showed that cmpd 31 possessed a better potency in inhibition of C4-2B CRPC cell growth (with IC_50_ of 1.51 μM) than XY018 or GSK805 (IC_50_ of 3.8 and 6.34, respectively) (Appendix A). Our colony formation assay demonstrated that like XY018, cmpd 31 was very effective in inhibition of the CRPC cell survival (Figure 1C,D and Appendix A). Similar to XY018, cmpd 31 strongly suppressed the protein expression of RORγ, AR, AR variants and Myc in C4-2B cells (Figure 1E). While the levels of cleaved PARP were markedly upregulated by cmpd 31 and XY018, anti-apoptosis protein Bcl-2 was downregulated by the treatment, indicating that both of them promoted CRPC cell apoptosis (Figure 1E). Furthermore, we found that the migration and invasion capacities of C4-2B cells were also dramatically impaired by XY018 and cmpd 31 (Figure 1F). In further support of our hypothesis that RORγ plays a role in control of CRPC cell invasion, we observed that RORγ knockdown strongly inhibited the migration and invasion of C4-2B cells (Appendix A). Together, we found that like the other amide derivatives XY018 and XY101, cmpd 31 possesses potent activities in inhibition of CRPC cell proliferation, survival, migration, and invasion.

### 2.2. Orally Administered RORγ Antagonists/Inverse Agonists Potently Inhibit Growth of PDX Tumors

The remarkable activity of cmpd 31 in inhibition of CRPC cell growth prompted us to examine its anti-tumor potency. As shown in Figure 2A, at a dose of 5 mg/kg (i.p.) cmpd 31 strongly inhibited the growth of C4-2B xenograft tumors with an efficacy similar to that of XY018 and XY101 as we reported previously [18,22] (Figure 2A). In order to evaluate their therapeutic effect in a more clinically relevant setting, we tested the efficacy of oral administration of antagonists XY018, cmpd 31, and GSK805 at two doses (20 mg/kg or 40 mg/kg) in animals bearing an AR-positive CRPC PDX model LuCaP 35CR [26,27]. The growth of LuCaP-35CR xenografts was significantly repressed after 40 days of treatment with all three RORγ antagonists in a dose-dependent manner. Consistent with their effectiveness in the cell culture, XY018 and cmpd 31 displayed strong anti-tumor potencies which were much higher than that of GSK805 (Figure 1B and Appendix A). As demonstrated in our previous studies, the three RORγ antagonists suppressed tumor growth without any significant effect on the animal body weight (Appendix A). Our IHC analysis of tumor tissues showed that Ki-67 positive cells were drastically decreased while cleaved caspase-3 positive cells were significantly increased in RORγ antagonist-treated tumors, indicating that ROR-γ inhibition suppressed tumor cell proliferation and induced cell apoptosis in vivo (Figure 2C–E). Thus, these data demonstrated that RORγ antagonists, particularly cmpd 31 and XY018, exhibited a potent anti-tumor activity in both cell line-derived tumors and in PDX models when administered orally.

### 2.3. RORγ Antagonists/Inverse Agonists Suppress the Expression of Gene Programs Linked to Tumor Aggressiveness

To examine the mechanism of the RORγ antagonists in inhibition of CRPC, we performed RNA-seq analysis with C4-2B cells treated with the antagonists. As shown in Figure 3A, treatments by the three RORγ antagonists resulted in a strong overlap of downregulated genes (36.6% of XY018, 26.8% of cmpd 31, and 39.8% of GSK805). Intriguingly, the overlapped, downregulated genes were strongly enriched in a gene set called SEG1 (subtype-enriched genes 1) and cmpd 31 appeared to most strongly suppress SEG1 genes among the three antagonists (Figure 3B). SEG1 was previously identified in a tumor subtyping study through meta-analysis of gene profiles of PCa tumors [28]. SEG1 genes are highly expressed in PCa subtype 1 (PCS1) tumors that exhibits the highest risk of progression to aggressive or lethal form of PCa when compared to tumors enriched for SEG2 or SEG3. Our gene set enrichment analysis (GSEA) also revealed that a significant inhibition of PCS1 gene programs by the three RORγ antagonists (Appendix A). Furthermore, our qRT-PCR analysis of a 37-gene panel that possesses the subtyping power [28] verified the selective downregulation of PCS1 signature genes revealed by our RNA-seq analysis (Figure 3C). To assess the impact on the gene expressions in vivo, we treated mice bearing LuCaP-35CR tumors with 20 mg/kg/d of XY018 or cmpd 31 for 14 days and then harvested the tumors for qRT-PCR analysis. As shown in Figure 3D, the expression of PCS1-enriched genes—including STMN1, CCNB1, CDC6, CDKN3, TPX2, KIF11, HMMR, MKI67, and KNTC1—was significantly inhibited in tumors treated with cmpd 31, and to less extent in tumors treated with XY018. We also performed GSEA with seven different sets of PCa-relevant pathway signature genes and found that in addition to AR targets, programs of aggressive AR variant, stemness and NEPC were significantly enriched in the CRPC cells (Appendix A). Since cmpd 31 and XY018 potently inhibited CRPC cell migration and invasion, we explored further whether RORγ is involved in EMT process. Our GSEA analysis revealed that indeed EMT-related genes were significantly suppressed by the compounds in C4-2B cells (Figure 3E). Furthermore, we confirmed by qRT-PCR and western blotting that the expression of four EMT-related signature genes PBK, NEK2, PBK, and FN1, which displayed the largest fold change in RNA-seq after treatment, was strongly decreased by the antagonists (Figure 3F,G). We also found that MMP-1 and MMP-2 proteins, major matrix metallopeptidases involved in invasion, were also strongly suppressed by XY018 and cmpd 31 (Figure 3G). Together, these results suggest that RORγ antagonists cmpd 31 and XY018 strongly suppressed the expression of aggressive gene programs of PCa, including genes involved in EMT and invasion.

### 2.4. PBK Is a Downstream Target of RORγ

The pro-oncogenic serine/threonine kinase PBK was recently shown being overexpressed in metastatic PCa tumors and regulated by AR [9]. Given that RORγ controls AR expression and function [18,22], we asked the question whether PBK is also under the control by RORγ. Thus, we first performed knockdown to silence the expression of RORγ. Indeed, knockdown of RORγ expression reduced not only the expression of AR but also PBK mRNA and protein in both C4-2B and 22Rv1 cells (Figure 4A–C). We also found that CRISPR-Cas9 knockout of RORγ gene strongly reduced the expression of PBK (Figure 4D) whereas RORγ overexpression resulted in an increased level of PBK protein (Figure 4E). Interestingly, in both cell number counting and CellTiter-Glo assay, RORγ overexpressed CRPC cells displayed significantly decreased sensitivity to the PBK inhibitor when compared to vector control cells (Figure 4F,G). These results clearly demonstrated that RORγ controls the expression and function PBK in CRPC cells.

### 2.5. PBK Interacts With RORγ and Modulates RORγ Protein Stability

To further explore the functional interactions between PBK and RORγ, we treated C4-2B cells with increasing doses of the PBK inhibitor (PBKi) and measured the expression of AR, RORγ, and PBK. Interestingly, we found that the protein levels of RORγ, AR, and PBK were significantly decreased by the PBKi in a dose-dependent manner (Figure 5A) whereas the mRNA level of RORγ/RORC was not significantly altered (Figure 5B). These results prompted us to speculate that PBK might interact with RORγ and thereby stabilize RORγ protein. We thus performed co-immunoprecipitation (IP) experiment and found that indeed PBK protein interacted with RORγ protein (Figure 5C). To assess the role of PBK on RORγ protein stability, we first treated cells with PBKi or vehicle for 48 h and then with cycloheximide (CHX) to inhibit new protein synthesis. Consistent with the previous study [9], AR protein was less stable in the presence of PBKi. Remarkably, RORγ protein in PBKi-treated cells also exhibited accelerated degradation (Figure 5D,E). In support of the hypothesis that proteasome-mediated degradation contributes to the PBKi-induced degradation of RORγ, we found that the decrease of RORγ protein by PBKi treatment was attenuated by proteasome inhibitor MG132 (Figure 5F). We also observed that PBKi treatment resulted in a marked increase of polyubiquitinated RORγ in the CRPC cells (Figure 5G). Collectively, our data revealed a novel feedforward loop between PBK and RORγ where RORγ activates the expression of PBK that in turn stabilizes RORγ and drives PCa progression.

### 2.6. Dual Inhibition of PBK and RORγ Synergistically Inhibits CRPC Cell Survival and Invasion and AR Signaling

Our identification of RORγ-PBK as a novel regulatory axis prompted us to examine the consequences of simultaneous targeting of the two proteins. Indeed, treatment with PBKi at 0.4 μM was sufficient to sensitize C4-2B cells to RORγ antagonist cmpd 31 as suggested by an approximately 4-fold decrease (from 0.156 to 0.039 μM) of cmpd 31 concentrations needed in suppression of C4-2B colony formation to close to 140 colonies (Figure 6A and Appendix A). Similar PBKi-induced sensitization to the RORγ antagonist was observed with 22Rv1 cells (Appendix A). Moreover, our transwell assays showed that dual inhibition of PBK and RORγ synergistically suppressed the migration and invasion of C4-2B cells (Figure 6B). Finally, we found that a combined treatment of the CRPC cells with both the PBKi and the RORγ antagonist elicited a robust and synergistic inhibition of the expression of both AR full-length and AR-V7 proteins as well as RORγ in 22Rv1 cells (Figure 6C). Our qRT-PCR analysis also demonstrated that the combined treatment synergistically inhibited the expression of AR signaling-regulated genes including those involved in cell proliferation and survival (e.g., PLK1, CCNB1, BIRC5, CENPF, CDC6, and CDC45) (Figure 6D). Together, our results suggest that targeting AR by PBK and RORγ dual inhibition might be a promising strategy for CRPC treatment.

## 3. Discussion

Hyperactive AR signaling due to AR gene alterations and AR variants is a well- known major driver of PCa progression to mCRPCs [29,30]. We previously identified nuclear receptor RORγ as a key upstream regulator of AR [18]. Based on the strong anti-tumor efficacy of RORγ antagonists XY018, XY101, and SR2211 in PCa cell line-derived xenograft models, we proposed that targeting RORγ can be an efficacious strategy to treat advanced CRPCs [18,22]. In this study, we report that several antagonists including XY018 and cmpd 31, when administered orally, displayed dose-dependent anti-tumor activities in a PDX model of CRPC. Our molecular analyses revealed that the RORγ antagonists could strongly inhibit the expression of not only AR and AR variants but also gene programs that are associated with an aggressive subtype of PCa, including genes involved in EMT and cell growth and proliferation. Indeed, our further experiments showed that the antagonists inhibit CRPC cell migration and invasion, which is consistent with our previous results that small molecule antagonists of RORγ can inhibit tumor metastasis [18,24]. Moreover, consistent with our previous studies, oral dosing of animals carrying the PDX tumors with the antagonists does not cause overt impact to the host animals. Together, our results clearly indicate the therapeutic value of targeting RORγ in CRPC.

In our attempt of revealing the major gene programs perturbed by the RORγ antagonists, we identified several major EMT regulators such as PBK, FN1, NEK2, and PTTG1 that are strongly inhibited by the antagonists. Indeed, the serine/threonine kinase PBK was of particular interest to us. Recently, several studies showed that PBK is overexpressed in multiple cancer types including prostate cancer [9,31,32,33,34]. In a proteomic study, Warren et al. found that PBK is upregulated by AR and that PBK in turn directly interacts with the N-terminus of AR to stabilize AR protein, thus representing a novel feed-forward mechanism in control of AR signaling [9]. Here, we provide evidence that PBK also interacts with RORγ and stabilizes RORγ protein. Therefore, our data demonstrate an interplay between RORγ and PBK via another feed-forward mechanism operating in mCRPC cells that is interlocked with the PBK-AR loop. With the role of RORγ in control of AR and PBK expression, we propose a model (Figure 6E) where overexpressed RORγ stimulates the expression of AR that in turn upregulates PBK expression. Increased PBK interacts with and stabilizes RORγ and AR, that in turn reinforces PBK activation and further amplifies AR signaling in mCRPC. One prediction from this model is that disruption of the loops would render the CRPC cells highly sensitive to therapeutics targeting RORγ and PBK (Figure 6E). Indeed, we observed that dual inhibition RORγ and PBK synergistically suppressed AR signaling and inhibited the growth and survival of CRPC cells. Currently, little is known about regulations of RORγ gene expression or its protein stability in cancer cells. On the other hand, for T cell-specific RORγt, there have been several elegant studies on its phosphorylation and ubiquitination regulation by IκBα kinase, ITCH, TRAF5, and USPs as a mechanism of its protein stability or function [35,36,37,38,39]. For example, deubiquitinase DUBA can suppress RORγt protein stability in TGF-β stimulated Th17 cells [40]. Whether similar or different regulation mechanisms and players operate in prostate cancer cells await to be determined. Our results that PBK kinase inhibitor increases RORγ ubiquitination suggest that phosphorylation of RORγ at specific sites likely blocks its ubiquitination. Given the prominent interplays among the three druggable tumorigenic factors—namely RORγ, PBK, and AR—further investigations are warranted to define the role of phosphorylation and ubiquitination in control of RORγ functions in prostate cancer and to elucidate the mechanisms underlying the interactions among the three proteins.

## 4. Materials and Methods

### 4.1. Cell Culture and siRNA Transfection

C4-2B cells were from UroCor Inc. 22Rv1 cells were from ATCC. They were cultured in RPMI 1640 (Corning, New York, NY, USA cat. no. 10-040-CM) containing 10% Fetal Bovine Serum (FBS) (Thermo Fisher Scientific, Waltham, MA, USA cat. no. A3160502). To mimic the castration-resistant prostate cancer (CRPC) condition, C4-2B cells were cultured in phenol red-free RPMI 1640 (Thermo Fisher Scientific) supplemented with 9% charcoal-dextran-stripped (cds) FBS, plus 1% FBS for experiment. All cells were grown in a humidified incubator with 5% CO_2_ at 37 °C. siRNAs were purchased from Dharmacon (Cambridge, UK). The siRNA sequences targeting RORγ gene RORC were CGAGGATGAGATTGCCCTCTA for siRORC-1, and CACCTCACAAATTGAAGTGAT for siRORC-2. siRNA sequence CAGTCGCGTTTGCGACTGG was used as non-targeting control. Transfection of siRNAs was performed with OptiMEM (Invitrogen, Carlsbad, CA, USA cat. no. 11058021) and Dharmafect1 (Dharmacon, Cambridge, UK cat. no. T-2001-02), following the manufacturer’s instructions.

### 4.2. Chemicals and Reagents

SR2211 was purchased from TOCRIS (Minneapolis, MN, USA cat. no. 4869). XY018, Cmpd 31 and GSK805 were synthesized and purified to >98–99% purity by WuXi AppTec. PBK/TOPK inhibitor HI-TOPK-032 was from Sigma (St. Louis, MO, USA cat. no. SML0796). Other chemicals were purchased from Thermo Fisher Scientific.

### 4.3. Cell Viability and Growth Assay

For cell viability, cells were seeded to 96-well plates at 1500/well density. Indicated concentrations of compounds were added to cells after 24 h. After 4 days of incubation, cell viability was measured using Cell-Titer GLO reagents (Promega, WI, USA cat. no. G9243) per manufacturer instructions. For cell growth assay, cells were seeded in 6-well plates at a density of 2 × 10^5^ per well and transfected with siRNA or treated with compounds as indicated. Total cell number were counted with a Coulter counter. For colony formation assay, 500 cells were seeded in a well of 6-well plate and cultured for 14 days. Then cell colonies were stained with Fix/Stain buffer (1 × PBS containing 0.05% Crystal Violet, 1% Formaldehyde, 1% Methanol) for 20 min at room temperature (RT) and washed with H_2_O five times and air-dried overnight. Cell colonies were then scanned and counted.

### 4.4. Western Blotting and qRT-PCR

Cells were lysed with RIPA lysis buffer (150 mM NaCl, 1% NP40, 0.5 % Sodium deoxycholate, 0.1 % SDS, 25 mM Tris-HCl pH7.4, 1 mM PMSF). Protein concentrations were measured and adjusted using DC™ Protein Assay Kit I (Bio-Rad, Hercules, CA, USA, cat. no. 5000111). Then proteins were separated by 10% SDS-PAGE and transferred onto PVDF membranes, which were then blocked by 5% fat-free milk. Membranes were incubated with indicated primary antibodies at 4 °C overnight and then subjected to second antibody (1:2000) incubation at RT for 1 h. Antibody-recognized proteins were visualized using ChemiDocTM MP imaging system (Bio-Rad) after incubation with HRP substrate (Millipore, San Francisco, NJ, USA, cat. no. WBLUR0500). Antibodies used in this study are described in Appendix A. Total RNA was isolated from cells in 6-well plates or from xenograft tumors with TRIzol reagent (Invitrogen, CA, USA, cat. no. 15596018). One µg of total RNA was reverse-transcribed to cDNA using qScript cDNA SuperMix (Quantabio, MA, USA, cat. no. 95048). qRT-PCR was performed as previously described [41]. Briefly, cDNAs were mixed with SYBR Green master mix (Bimake, Houston, TX, USA, B21202) and gene specific primers. The PCR was run on a CFX96 connect Real-Time PCR system (Bio-Rad). GAPDH gene transcript was used for normalization. The 2^−ΔΔCT^ method was used to obtain the relative quantifications. Experiments were repeated three times. The primers are listed in Appendix A.

### 4.5. Immunohistochemistry (IHC)

Briefly, dissected tumor tissues were fixed in 10% neutral buffered formalin overnight and then embedded in paraffin and cut into 4μm-thick sections. The sections were deparaffinized with xylene and rehydrated with gradient alcohol before antigen retrieval was performed in 0.01 M citrate buffer at just below boiling temperature for 10 min. Endogenous peroxidase activity was quenched with 3% H_2_O_2_ and then 10% goat serum were used for eliminating non-specific binding. Subsequently, tissue sections were incubated overnight at 4 °C with anti-Ki-67 monoclonal antibody (cat. no. 14-5698-82; Thermo Fisher Scientific) or anti-cleaved-Caspase-3 monoclonal antibody (Cell Signaling Technology, MA, USA, cat. no. 9664;), at 1:1000 and 1:500 dilutions, respectively. Then the slides were washed and incubated with biotin-conjugated secondary antibodies for 30 min followed by incubation with ABC reagents in the Vectastain Elite ABC kits (Vector Laboratories, CA, USA, cat. no. PK-7200) and counterstaining with hematoxylin, as previously described [42]. Images were obtained using an Olympus microscope with DP Controller software (Olympus, Waltham, MA, USA, version 3.2.1.276).

### 4.6. Transwell Assay

For Transwell assays, C4-2B cells were pretreated with siRNAs, compounds or RORγ-V5 lentivirus for 48 h and then detached into single-cell suspensions. Cells (1 × 10^5^) were then resuspended in RPMI medium containing 0.1% FBS and placed into the upper chamber of Transwell inserts with 8 μm pores (Corning, NY, USA, cat. no. CLS3422) for migration assays or 2 mg/mL Matrigel-coated Transwell inserts for invasion assays. The bottom wells contained 500 µL 10% FBS-RPMI as a chemo-attractant. After 24 h (for migration assay) or 48 h (for invasion assay), cells on the upper surfaces of the inserts were removed using a cotton swab. Migrated cells on the lower surface of the membrane were fixed in 4% formaldehyde and stained with 0.1% crystal violet for 15 min. Cells were counted under microscopy for ten random fields.

### 4.7. Lentivirus Production, Cell Infection, co-IP, and Ubiquitin Pull Down

Human RORγ cDNA was cloned into pLX304-V5 plasmid as previously described [18]. sgRNA sequences for RORγ gene RORC is GTGGGGCTGTCCAAGTGACC and sgGFP is GGGCGAGGAGCTGTTCACCG. sgRNA sequences were cloned into lentiCRISPR V2 vector. For lentivirus production, 239T cells (4 × 10^6^ per 100 mm dish) were co-transfected with pLX304-V5-RORγ or lentiCRISPR V2 and packaging plasmids psPAX2 and pMD2.G using lipofectamine 2000 overnight. Then medium was replaced with fresh medium. After 48 h, virus-containing supernatants were harvested and stored in −80 °C freezers before use. Cells (2 × 10^5^) in 6-well were infected with the lentiviruses for expressing V5-RORγ or empty vector by incubation with 1 mL of the supernatant supplemented with 10 ng polybrene for 6 h. For co-IP assay, magnetic beads were precoated with 4 μg indicated antibodies at 4 °C overnight. Cells were lysed with lysis buffer (10 mM HEPES pH7.9, 10 mM KCl, 0.1 mM EDTA, 0.4% NP-40 and protease inhibitor cocktail) for 30 min at 4 °C. Then the lysates were centrifuged at 15,000× *g* for 1 min to collect nuclei pellets that were then incubated in extraction buffer (20 mM HEPES pH7.9, 0.4 M NaCl, 1 mM EDTA and protease inhibitor cocktail) for 15 min to extract nuclear proteins. After protein concentrations were adjusted, 1% nuclear extracts were kept as input. The remaining extracts were divided into two equal groups: one group was incubated with anti-V5 antibody-coated magnetic beads and the other was incubated with IgG-coated magnetic beads. After incubation at 4 °C overnight, the beads were washed using washing buffer (50 mM Tris-HCl pH7.5, 200 mM NaCl, 5 mM EDTA, 1% Triton) for 5 times and eluted in 2 × SDS protein loading buffer (0.125 M Tris-HCl pH 6.8, 4% SDS, 20% Glycerol, 0.004% Bromophenol blue, 10% 2-mercaptoethanol). Then the eluted proteins were subjected to Western blotting. For ubiquitin pull down assay, cells were lysed with NP40 lysis buffer (150 mM NaCl, 50 mM Tril-HCl pH8.0, 1% NP-40 and protease inhibitor cocktail) for 30 min at 4 °C. Then cell lysates were centrifuged at 12,000× *g* for 10 min to collect supernatants. Equal amounts of supernatants were incubated with anti-V5 Ab-coated or IgG coated magnetic beads. The subsequent steps were the same as Co-IP assay.

### 4.8. Xenograft Tumor Models and Treatments

All animal experiments were approved by the Institutional Animal Care and Use Committee of the University of California, Davis. Four-weeks-old male NOD-SCID mice (NOD.Cg-Prkdc<scid>/J, Envigo) were castrated and two weeks later subcutaneously implanted with LuCaP 35CR [26] tumor tissues by trocar insertion. When the tumor volume was approximately 100 mm^3^, the mice were randomized and then treated orally (p.o.) with 100 µL of either vehicle or RORγ antagonist XY018, Cmpd 31 or GSK805 (in a formulation of 15% Cremophor EL, Calbiochem, 82.5% PBS and 2.5% dimethyl sulfoxide (DMSO)) for five times per week. Tumor growth was monitored by calipers, and volume was calculated with the equation V = 0.5 × (length × width2). Mice carrying C4-2B tumors were established and treated in the same way as with LuCaP 35CR model. Mouse body weight during the study was also monitored. At the end of the studies, mice were euthanized, and tumors were dissected, weighed, and processed for further analyses.

### 4.9. RNA-Seq and Bioinformatics Analysis

C4-2B cells were treated with RORγ antagonists XY018 (5 µM), Cmpd 31 (5 µM), GSK805 (5 µM), or vehicle for 48 h before RNA isolation. RNA-seq libraries from 1 µg total RNA were prepared and validated as previously described [18]. Sequencing was performed on an Illumina HiSeq 2000 sequencer at BGI Tech (Hong Kong, China). The FASTQ files were aligned to the reference human-genome assembly (GRCh37/hg19, released Feb 2009) with BWA and Bowtie software (access date 2 January 2019). The cufflinks package [43] was subsequently applied for differential transcript expression analysis. To avoid spurious fold levels resulting from low expression values, only those genes with expression RPKM (reads per kilobase per million mapped reads) or FPKM (fragments per kilobase of exon model per million mapped reads) values no less than 0.5 for all cells were included. Heatmaps were generated on log transformation of normalized expression using the heatmap package in R language (https://cran.r-project.org/web/packages/pheatmap/index.html, access date 4 January 2019). GSEA was conducted using the Java desktop software (http://software.broadinstitute.org/gsea/index.jsp, access date 19 September 2019), as described previously [44]. Genes were ranked according to the shrunken limma log2 fold changes. The GSEA tool was used in ‘pre-ranked’ mode with all default parameters. Previous reported PCS1-subtype enriched genes, prostate cancer pathway activation signatures [28], and epithelial–mesenchymal transition (EMT)-related genes [45] were used in the GSEA analysis.

### 4.10. Statistical Analysis

Cell culture–based experiments were performed at least three times, with assay points triplicated, as indicated. The data are presented as mean ± SD from three independent experiments. Statistical analysis was performed using one-way ANOVA for qRT-PCR analysis, tumor xenografts, quantitative analysis of IHC data, wound healing, and transwell assay. Statistical significances were considered when were less than 0.05.

## 5. Conclusions

In conclusion, our study revealed RORγ-controlled aggressive gene programs and the interlocked feedforward loops as a new mechanism in hyperactive AR signaling in mCRPC, and further demonstrated the therapeutic efficacy of ROR-γ antagonists when orally administered in a clinically relevant PDX model. Thus, our study highlights the potential value of the RORγ antagonists for further development as attractive therapeutics for mCRPC.

## Figures and Tables

**Figure 1 cancers-13-01672-f001:**
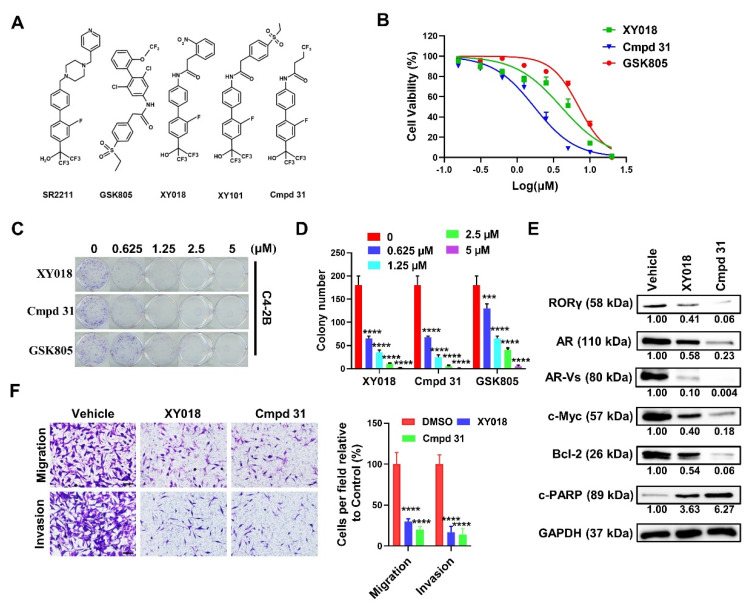
RORγ antagonists potently inhibit CRPC cell proliferation, migration and invasion. (**A**) Chemical structures of the RORγ antagonist compounds. (**B**) C4−2B cells seeded to 96 well plates were treated with indicated compounds for 96 h before cell viability was measured. (**C**,**D**) C4-2B cells were seeded to 6-well plates at 500/well and treated with different doses of indicated compounds for 2 weeks. The cell colonies were then stained and counted. (**E**) C4−2B cells were treated with 5 µM of indicated compounds for 72 h before processed for Western blotting. Density of each band was measured by ImageJ software and normalized with that of GAPDH. Data were presented as fold changes relative to the density of vehicle. (**F**) C4−2B cells were treated with 5 µM of XY018 or Cmpd 31 for 48 h before being subjected to migration and invasion assays. After incubation for 24 h (for migration) or 48 h (for invasion), migrated or invaded cells were fixed and stained. Scale bar, 200 µm. Cells were counted under microscopy for 5 random fields. Results are presented as mean ± SD. *** *p* < 0.001, **** *p* < 0.0001.

**Figure 2 cancers-13-01672-f002:**
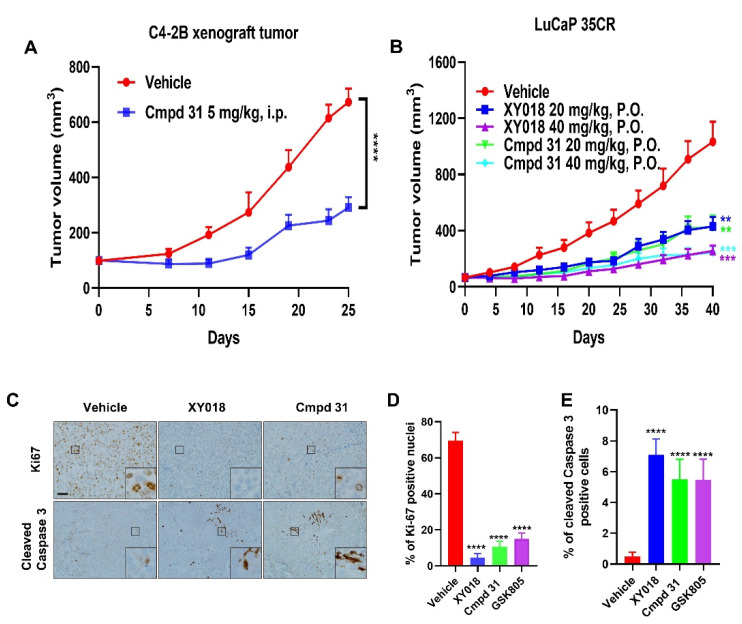
Orally administered RORγ antagonists exhibit strong anti-tumor activities. (**A**) C4−2B cells were subcutaneously xenografted on the flanks of NOD−SCID mice. When tumors reached 100 mm^3^, mice were divided into two groups (*n* = 8 tumors per group) and treated with vehicle or 5 mg/kg cmpd 31 (i.p.) five times per week for 25 days. Tumor volumes were monitored. (**B**) Mice with LuCaP-35CR PDX tumors were treated orally with RORγ antagonists Cmpd 31 and XY018 (20 mg/kg or 40 mg/kg) or vehicle (*n* = 8 tumors per group), five times per week. Tumor volumes were monitored. (**C**) Representative images from Ki−67 and cleaved−Caspase−3 immunohistochemistry of tumors from mice treated with 40 mg/kg of Cmpd 31, XY018, or vehicle. Scale bar: 50 µm. (**D**,**E**) Quantitative analysis of anti-Ki-67 positive nuclei or anti-cleaved caspase 3 stained cells in LuCaP−35CR tumors. The percentage of positive nuclei or cells were calculated by dividing the number of positive nuclei or cells by the number of total nuclei or cells per visual field. Results are presented as mean ± SD. ** *p* < 0.01, *** *p* < 0.001, **** *p* < 0.0001.

**Figure 3 cancers-13-01672-f003:**
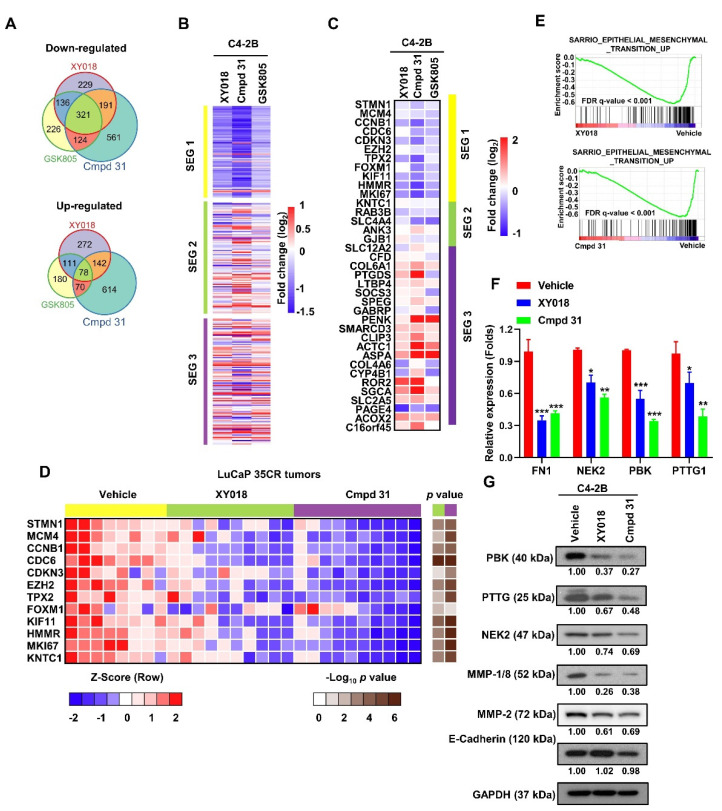
RORγ antagonists suppress the expression of PCS1 PCa subtype signature genes and EMT genes in PCa cells and PDX tumors. (**A**) Venn diagrams display the overlaps of up− or downregulated genes (>2−fold change) detected by RNA−seq in C4−2B cells treated with 5 µM of XY018, cmpd 31 or GSK805 for 48 h, when compared to vehicle. (**B**) Heatmap depicting the fold changes of PCS1 subtype-enriched genes (SEG 1), PCS2−enriched genes (SEG 2) and PCS3−enriched genes (SEG 3) in C4−2B cells after 48 h treatment with 5 µM of RORγ antagonists detected by RNA−seq (genes with mRNA FPKM ≤ 0.1 were excluded). (**C**) qRT-PCR analysis of the 37 SEG genes selected by You et al. in C4−2B cells treated with 5 µM of RORγ antagonists for 48 h. Results were displayed in heatmap. (**D**) Mice with LuCaP−35CR PDX tumors were treated via i.p. with 20 mg/kg/d of XY018, cmpd 31 or vehicle (*n* = 8−10 tumors per group) for 14 days. Tumor tissue was subjected to qRT−PCR analysis of PCS1−enriched genes. Results were displayed in heatmap. The *p*−value was determined using unpaired Student’s *t*−test (treatment vs control) with a two−tailed distribution. (**E**) GSEA of EMT−associated gene signatures in C4−2B cells treated with 5 µM of XY018 (top), Cmpd 31 (bottom). (**F**,**G**) qRT−PCR and western blot analysis of EMT associated genes or proteins in C4-2B cells treated with 5 µM of the antagonists for 48 h. Density of bands was measured and presented as in Figure 1. Results are presented as mean ± SD. * *p* < 0.05, ** *p* < 0.01, *** *p* < 0.001.

**Figure 4 cancers-13-01672-f004:**
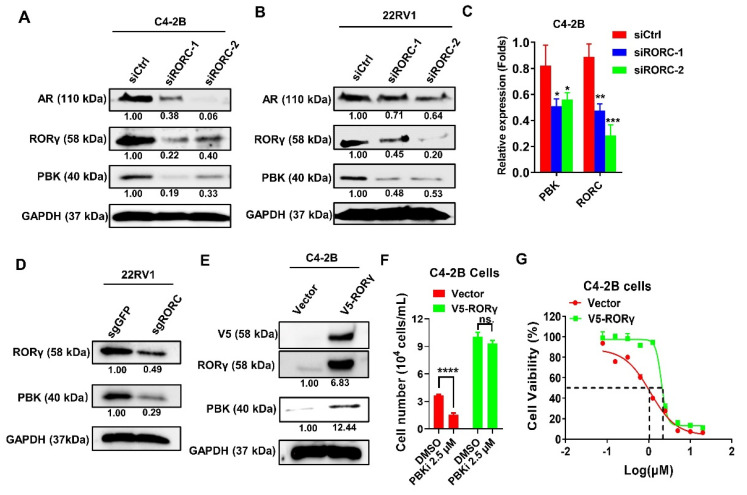
PDZ-binding kinase (PBK) is a downstream gene regulated by RORγ. (**A**,**B**) Western blotting analysis of C4−2B cells and 22Rv1 cells transfected with RORC siRNAs for 72 h. (**C**) qRT−PCR analysis of PBK and RORC mRNAs in C4−2B cells transfected with the siRNAs. (**D**,**E**) Western blotting of 22Rv1 or C4−2B cells infected with lentiviruses expressing CRISPR−Cas9 and sgRNA−RORC (**D**) or V5−RORγ (**E**) respectively or the control viruses for 72 h. (**F**) C4−2B cells were infected with lentiviruses expressing V5−RORγ or the control for 24 h and then treated with 2.5 μM PBKi for another 72 h before cell number was counted. (**G**) C4−2B cells were infected with lentiviruses expressing V5−RORγ or empty vector for 24 h and then detached and seeded to 96-well plates at 1500 cells per well. After 24 h recovery, cells were treated with increasing doses of PBKi for another 72 h before subject to cell viability measurements. Density of bands was measured and presented as in Figure 1. Results are presented as mean ± SD. * *p* < 0.05, ** *p* < 0.01, *** *p* < 0.001, **** *p* < 0.0001.

**Figure 5 cancers-13-01672-f005:**
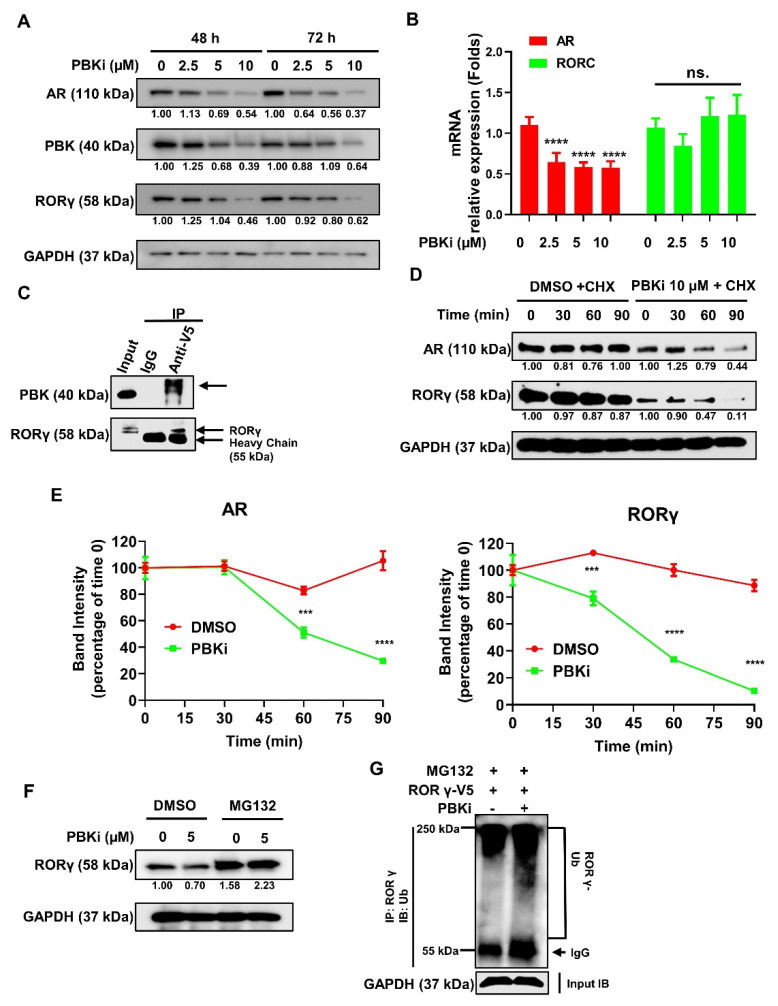
PBK interacts with RORγ and regulates its stability in CRPC cells. (**A**) Western blotting analysis of C4−2B cells treated with indicated concentrations of PBKi for 48 h and 72 h. (**B**) qRT−PCR analysis of AR and RORC mRNAs in C4-2B cells treated for 72 h treatment with PBKi. (**C**) co−IP analysis of C4−2B cells expressing V5−RORγ. Cell lysates were first pulled down by anti−V5 or control IgG antibody. The co−IPed complexes were then subject to western blotting with anti−PBK or RORγ antibody. (**D**) Western blotting of C4-2B cells incubated with 5 μM PBKi for 48 h and then treated with 50 μM CHX for indicated times. (**E**) The band intensity of each treatment as in D was analyzed using ImagJ and normalized to GAPDH. The percentage of band intensity was calculated by referring to time 0. (**F**) Western blotting of V5−RORγ−expressing C4-2B cells first treated with 5 μM PBKi for 48 h and then treated with 10 μM MG132 for 3 h. (**G**) RORγ protein was first pulled down by RORγ antibody and then ubiquitin levels of RORγ in the complexes were detected by anti-ubiquitin antibody. Density of bands was measured and presented as in Figure 1. Results are presented as mean ± SD. *** *p* < 0.001, **** *p* < 0.0001.

**Figure 6 cancers-13-01672-f006:**
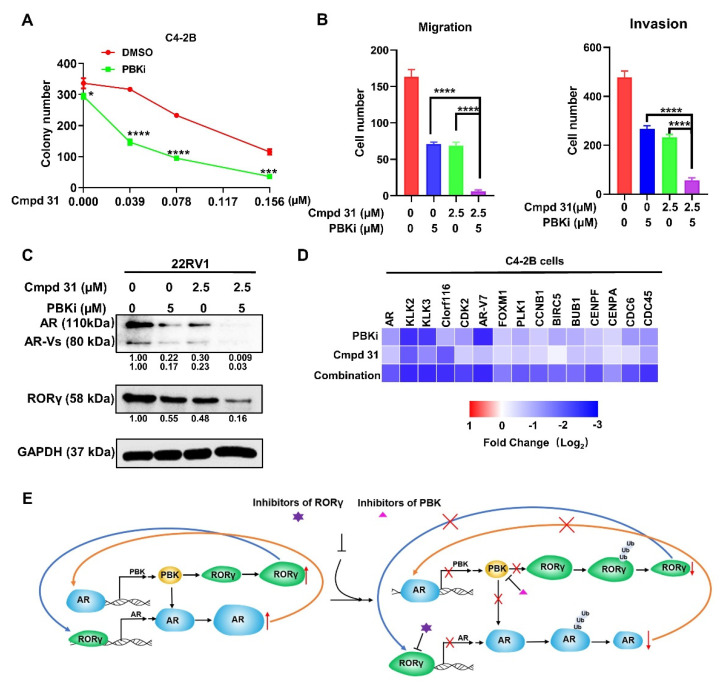
Dual inhibition of PBK and RORγ displays a strong synergistic effect in killing CRPC cells via targeting AR protein. (**A**) colony formation of C4−2B cells incubated with indicated PBKi, Cmpd 31, DMSO or combination for 10 days. (**B**) C4−2B cells were pretreated with PBKi, Cmpd 31 or combination as indicated, for 48 h before subjected to migration and invasion assay. After 24 h (for migration assay) or 48 h (for invasion assay), migrated or invasive cells were fixed and stained. Scale bar, 200 µm. Cells were counted under microscopy for 5 fields. (**C**) Western blotting of 22Rv1 cells treated with PBKi, Cmpd 31 or combination as indicated for 72 h. Density of bands was measured and presented as in Figure 1. (**D**) Heatmap display of log2 fold change (relative to vehicle treatment) in expression of genes in C4−2B cells treated with PBKi, Cmpd 31 or combination as in C for 72 h, which was measured by qRT−PCR. Results are presented as mean ± SD. * *p* < 0.05, *** *p* < 0.001, **** *p* < 0.0001. (**E**) a schematic diagram of RORγ and PBK in driving the feed-forward loops in hyperactive AR signaling. Note: the relative size differences in the drawings of a given protein depict the effects of RORγ and PBK activities (left) and the changes induced by treatment with the inhibitors (right). Additional description can be found in Discussion.

## Data Availability

The data used to support this research are available from the corresponding author upon request.

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
