# Peer review of "Targeting Feedforward Loops Formed by Nuclear Receptor RORγ and Kinase PBK in mCRPC with Hyperactive AR Signaling"

_cancers, 2021, doi:10.3390/cancers13071672_

Round 1

Reviewer 1 Report

This is an interesting paper which provides a novel approach to the treatment of Metastatic castration-resistant prostate cancer, although I am wondering whether RAR-related orphan receptor gamma is the crucial discriminant of AR action. Overall, the manuscript is well thought out and written, andthe approaches are valid.  Please check abbreviations. Partial text editing is needed 

Author Response

We thank this reviewer very much for his/her comments that “This is an interesting paper which provides a novel approach to the treatment of Metastatic castration-resistant prostate cancer”, and that “Overall, the manuscript is well thought out and written, and the approaches are valid.”

Other comments:

although I am wondering whether RAR-related orphan receptor gamma is the crucial discriminant of AR action.

Response:

We agree with the assessment of this reviewer that RORγ is likely one of the crucial determinants/discriminants of AR action. We thus revised our descriptions or discussions accordingly to reflect this notion.

Please check abbreviations. Partial text editing is needed.

Response:

Yes, we have now checked/write out all abbreviations and edited the text of our manuscript.

Reviewer 2 Report

This is a methodologically-sound paper that depicts a positive feedback loop between ROR and PBK. The main limitation is that this is an in-vitro work only. It is not very novel as the same group has performed similar workds in the past.

I do not have any major comments.

There are a few language corrections that should be made; for instance:

1.  "the advanced forms of prostate cancer is to targeting drivers other than AR" - should be "the advanced forms of prostate cancer is to target....."

2.  "provide a strong rationale" - please soften the claim. 

3.  "is a downstream target of RORγ to exert the cellular effects" = 'is a downstream target of ROR that exterts the cellular effects".

4. "kinase signalings" - should be 'kinase signaling pathways".

5. "Currently, therapeutics are being developed targeting" - should be "....are being developed to target...."

Author Response

We thank this reviewer very much for his/her comment that “This is a methodologically-sound paper”.

Other comments:

The main limitation is that this is an in-vitro work only. It is not very novel as the same group has performed similar work in the past.

Response:

We agree that some part of our study was conducted with in vitro cell culture models, particularly those experiments for demonstration of the protein interactions and protein stability analysis. However, we also provided results that were obtained using tumor models, such as the regulation of the expression of aggressive tumor gene programs by inhibition of RORγ and the effects of the antagonists on tumor growth.

As to its novelty, although we agree that this study is not entirely novel, in our opinion, it has the following new findings/results that are previously not reporter by us or others:

1). Mutual regulations by RORγ, PBK and AR constitute novel feedforward loops that are inter-connected;

2). RORγ targeting by orally dosing with several small-molecule antagonists, including a previously uncharacterized one, are highly effective in blocking growth of multiple tumor models including PDX model;

3). Dual inhibition of RORγ and PBK synergistically diminishes hyperactive AR signaling.

We sincerely hope that the above assessment is shared by this reviewer.

There are a few language corrections that should be made; for instance:

  1. "the advanced forms of prostate cancer is to targeting drivers other than AR" - should be "the advanced forms of prostate cancer is to target....."

Response:

Thanks. We now made the correction.

  1. "provide a strong rationale" - please soften the claim.

Response:

Thanks. We now made the correction.

  1. "is a downstream target of RORγ to exert the cellular effects" = 'is a downstream target of ROR that exterts the cellular effects".

Response:

Thanks. We now made the correction.

  1. "kinase signalings" - should be 'kinase signaling pathways".

Response:

Thanks. We now made the correction.

  1. "Currently, therapeutics are being developed targeting" - should be "....are being developed to target...."

Response:

Thanks. We now made the correction.

Reviewer 3 Report

I read with interest the work titled Targeting Feedforward Loops Formed by Nuclear Receptor RORγ and Kinase PBK in mCRPC with Hyperactivating AR Signaling. The content of the work is of interest, however I must admit that I had difficulty in following the logic of the work.
Probably following a more canonical formulation could make the content of the work more usable and appreciable, which I repeat, is however interesting.
I reserve the right to re-evaluate the work after it has been reformulated according to the classic scheme of writing of scientific works.
In my opinion, the materials and methods section must also include a rewrite providing a more complete description of the method used, avoiding constant references to previous works by the group

Author Response

We thank this reviewer very much for his/her comment that “The content of the work is of interest”.

Other comments:

, however I must admit that I had difficulty in following the logic of the work.

Probably following a more canonical formulation could make the content of the work more usable and appreciable, which I repeat, is however interesting.

I reserve the right to re-evaluate the work after it has been reformulated according to the classic scheme of writing of scientific works.

In my opinion, the materials and methods section must also include a rewrite providing a more complete description of the method used, avoiding constant references to previous works by the group.

Response:

We are not entirely sure about “the logic of the work” and “following a more canonical formulation” referred to by this reviewer. Therefore, it is difficult for us to address those comments. For this, we apologize.

For the comment on M&M section, we have made extensive changes by providing details on the materials and methods and limiting our references to other previous studies.

Round 2

Reviewer 3 Report

I have no comments